# Lung Segmentation from Chest X-rays using Variational Data Imputation

**Raghavendra Selvan** [1]  **Erik B. Dam** [1 2]  **Nicki S. Detlefsen** [3]  **Sofus Rischel** [2]  **Kaining Sheng** [2]  **Mads Nielsen** [1 2]
**Akshay Pai** [1 2]

## Abstract

Pulmonary opacification is the inflammation in the lungs caused by many respiratory ailments, including the novel corona virus disease 2019 (COVID-19). Chest X-rays (CXRs) with such opacifications render regions of lungs imperceptible, making it difficult to perform automated image analysis on them. In this work, we focus on segmenting lungs from such abnormal CXRs as part of a pipeline aimed at automated risk scoring of COVID-19 from CXRs. We treat the high opacity regions as missing data and present a modified CNN-based image segmentation network that utilizes a deep generative model for data imputation. We train this model on normal CXRs with extensive data augmentation and demonstrate the usefulness of this model to extend to cases with extreme abnormalities. *

## 1. Introduction

Acute respiratory distress syndrome (ARDS) is characterized by rapid onset of inflammation in the lungs resulting in acute lung injury (Ware & Matthay, 2000). The extent of lung infection is often used as a marker for measuring the disease severity.

Imaging techniques are routinely employed to measure volume of lung infection due to ARDS (Bordley et al., 2004); this has also been attempted in detection of COVID-19 (Wong et al., 2020; Shi et al., 2020; Cohen et al., 2020). As chest X-rays (CXRs) are easier to obtain than computed tomography (CT) scans, they are more regularly used to perform early stage triaging of patients with ARDS and currently with COVID-19 symptoms. Obtaining accurate

---

[1]Department of Computer Science, University of Copenhagen, Copenhagen, Denmark [2]Cerebriu AS, Copenhagen, Denmark [3]Technical University of Denmark, Denmark. Correspondence to: Raghavendra Selvan <raghav@di.ku.dk>.

*Presented at the first Workshop on the Art of Learning with Missing Values (Artemiss) hosted by the $37^{th}$ International Conference on Machine Learning (ICML).* Copyright 2020 by the author(s).

*Trained models and the source code are available here: https://github.com/raghavian/lungVAE/

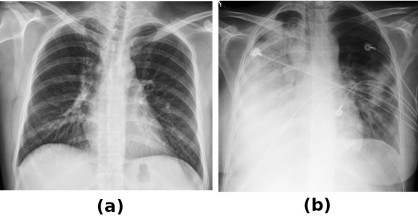

*Figure 1.* a) Normal chest X-ray showing the lungs clearly b) Abnormal CXR with high opacity where the right lung is hardly seen. Brighter regions are tissue-like as they attenuate X-rays whereas darker regions indicate presence of air, in this case inside the lungs.

segmentation of lung fields from CXRs is an essential first step in this process. However, extreme levels of opacification obfuscate large regions in the lungs making even manual segmentation of lungs difficult (Jacobi et al., 2020).

Prior to deep learning, automatic segmentation of lungs from CXRs was primarily based on active shape analysis (Xu et al., 2012) and deformable models (Candemir et al., 2013). With the advancement of fully convolutional neural (FCN) networks, CNN based methods have become state-of-the-art in various medical imaging tasks, including in lung segmentation tasks (Long et al., 2015; Ronneberger et al., 2015; Shin et al., 2016). However, most of these methods operate based on a strong (and commonly used) assumption that the out-of-sample/test data points are also from the same distribution as the training set (Wen et al., 2014). As a consequence, in lung segmentation tasks, segmentation models trained on CXRs with low opacification could fail to segment abnormal CXRs as their features can be vastly different as seen in Figure 1.

Two recent methods have focused on segmenting lungs from high opacity CXRs, to the best of our knowledge (Souza et al., 2019; Tang et al., 2019). In (Souza et al., 2019), initial segmentations are obtained from a patch classification network and refined further using a reconstruction network. However, this method requires a reasonable amount of labeled examples for the abnormal cases. In (Tang et al., 2019), the authors build on the capability of deep generative models to obtain realistic synthetic abnormal CXRs using adversarial training (Dai et al., 2018) and use these synthetic scans to train their segmentation model. This is a form of data augmentation, and it can also be limiting as the diversity of opacifications that can be realized using adversarial

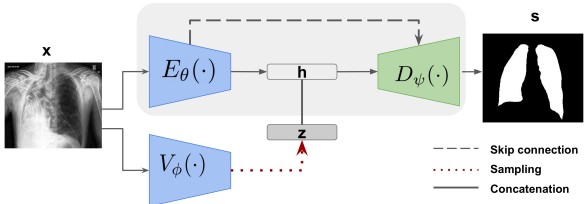

*Figure 2.* Overview of the proposed model with a variational encoder for data imputation, $V_\phi(\cdot)$ and a U-net type segmentation network with encoder $E_\theta(\cdot)$ and decoder $D_\psi(\cdot)$ (highlighted inside the grey box). The decoder is shared between the data imputation block and the segmentation network.

training are largely decided by the training samples. A thorough review of lung segmentation methods from CXRs is reported in (Candemir & Antani, 2019).

In this work, we aim to segment high opacity CXRs at test time by training primarily on normal CXRs. We treat this setting as dealing with incomplete data, as the training set does not contain high opacity images. Further, the opacification in CXRs itself is treated as the missing data that is to be inferred (Figure 1). While we also rely on specialized data augmentation similar to (Tang et al., 2019), presented in Section 2.2, we take up an alternative approach that builds on the strengths of deep latent variable generative models such as variational autoencoders (Kingma & Welling, 2014). We add a variational encoder to impute data by concatenating samples from the learnt latent space to a standard CNN based segmentation network, which is then jointly decoded to obtain the segmentations. We demonstrate the usefulness of the proposed approach by training on labeled examples of normal CXRs and testing on extreme cases of opacification.

## 2. Methods

Consider input images $\mathbf{x} \in \mathcal{X}$ and their corresponding segmentations $\mathbf{s} \in \mathcal{S}$, then the task of supervised image segmentation can be formulated as obtaining a mapping $f(\cdot) : \mathcal{X} \to \mathcal{S}$.

For a U-net type model (Ronneberger et al., 2015) the mapping function is composed of an encoder and decoder, such that $f(\cdot) = E_\theta(D_\psi(\cdot))$ where $E_\theta, D_\psi$ are encoder and decoder neural networks parameterised by $\theta$ and $\psi$ respectively, as shown in Figure 2.

### 2.1. Variational Data Imputation

Variational autoencoders have been used widely in generative settings as they can capture rich latent representations, which also make them a good fit for performing data imputation (Nazabal et al., 2018; Ham et al.). In the generative setting the optimization objective of a VAE is the evidence lower bound (ELBO) given by

$$\mathcal{L}_{VAE}(\mathbf{x}, \hat{\mathbf{x}}) = \mathcal{L}_{rec}(\mathbf{x}, \hat{\mathbf{x}}) + KL\big[q_\phi(\mathbf{z}|\mathbf{x})||p(\mathbf{z})\big] \quad (1)$$

where the first term, interpreted as reconstruction loss, is the negative expected log likelihood,

$$\mathcal{L}_{rec}(\mathbf{x}, \hat{\mathbf{x}}) = -\mathbb{E}_{q_{\mathbf{z}|\mathbf{x}}}[\log(p_\psi(\mathbf{x}|\mathbf{z}))]. \quad (2)$$

where $\hat{\mathbf{x}}$ is the reconstructed input. The second term is the KL divergence between the approximating variational density $q_\phi(\mathbf{z}|\mathbf{x}) = N(\mathbf{z}; \mu_\phi, \sigma_\phi^2)$ with the standard normal prior on the latent variable $p(\mathbf{z}) = \mathcal{N}(\mathbf{z}; 0, 1)$. The parameters of the axis aligned Gaussian $(\mu_\phi, \sigma_\phi^2)$ are predicted by the encoder neural network $V_\phi(\cdot)$ with parameters $\phi$.

In this work, the VAE is not used as an autoencoder but as a method to perform cross-domain mapping between the input $\mathcal{X}$ and the target segmentation $\mathcal{S}$ domains. This is in contrast with (Myronenko, 2018), where the VAE was used to reconstruct the input image to have a regularizing effect on the encoder layers. The proposed use of VAE bears similarities with the non-adversarial domain mapping work such as in (Hoshen & Wolf, 2018; Hoshen, 2018).

We introduce the latent random variable $\mathbf{z}$ to obtain low dimensional representations of the data, $\mathbf{x}$. We train the model with different augmentation strategies (Sec. 2.2) to learn a latent representation that can perform data imputation, handle missing data and possibly capture other task specific features such as shape information (Esser et al., 2018).

As depicted in Figure 2, the variational encoder, $V_\phi(\cdot)$, maps input images to a low dimensional latent space and samples from the latent space are concatenated to the output of the encoder of the segmentation network, $E_\theta(\cdot)$, depicted in Fig. 2. The decoder $D_\psi(\cdot)$ is shared between the U-net and the VAE such that they can jointly decode the segmentation $\mathbf{s}$, resulting in the following objective (Hoshen, 2018):

$$\mathcal{L}\big(\mathbf{s}, \hat{\mathbf{s}}\big) = \mathcal{L}_{rec}(\mathbf{s}, \hat{\mathbf{s}}) + KL\big[q_\phi(\mathbf{z}|\mathbf{x})||p(\mathbf{z})\big] \quad (3)$$

where the predicted segmentation, $\hat{\mathbf{s}}$, is obtained from the decoder:

$$\hat{\mathbf{s}} = D_\psi\big[E_\theta(\mathbf{x}) \ddagger V_\phi(\mathbf{x})\big] = D_\psi\big[\mathbf{h} \ddagger \mathbf{z}\big] \quad (4)$$

where $\ddagger$ is used to indicate concatenation, $\mathbf{h} = E_\theta(\mathbf{x})$ is the output of the U-net encoder and $\mathbf{z} \sim q_\phi(\mathbf{z}|\mathbf{x})$ is a sample from the latent space learnt by the variational encoder. Note that the objective in Eq. (3) is similar to the ELBO objective in Eq. (1) except for the reconstruction loss, which is the standard segmentation loss computed between $\hat{\mathbf{s}}$, the predicted segmentation and $\mathbf{s}$, the target segmentation. As the segmentation masks are binary we use binary cross entropy loss as the reconstruction loss. A reasonable interpretation of the objective in Eq. (3) is that the first term helps in segmentation while the second term has a regularisation effect (Kingma & Welling, 2014; Myronenko, 2018) and helps with data imputation.

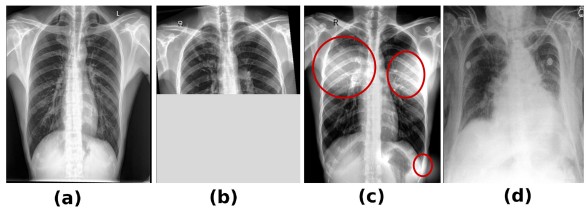

*Figure 3.* Chest X-rays with and without augmentation.
(a) No augmentation (b) With block masking (c) With diffused noise marked with red ellipses (d) Test image with high opacity

## 2.2. Augmentation strategies

Data augmentation strategies are now common practice when training deep learning models. They are primarily used to alleviate overfitting when labeled examples are scarce (Shorten & Khoshgoftaar, 2019). In self-supervised learning, data augmentation techniques are utilised to uncover expressive latent representations that could be useful in downstream tasks (Kolesnikov et al., 2019).

We use data augmentation extensively in this work. This is to simulate missing data instances such that the variational data imputation block can learn robust latent representations that can generalize well enough to high opacity CXRs at test time. We experiment with three types of data augmentations: 1) Standard 2) Block masking 3) Diffused noise.

The *standard* augmentation techniques used are random rotations, random horizontal and vertical flips. The *block masking* technique simply leaves out one half of the input image either horizontally or vertically (Fig. 3-b). Block masking simulates extreme opacifications where either entire lungs or large portions of it are missing due to opacification. This is similar to the class of random erasing techniques have been found to be useful in other image analysis tasks (Zhong et al., 2017). Finally, the *diffused noise* model is task specific to segmenting the high opacity in CXRs. We utilize a Strauss process realization (Descombes & Zerubia, 2002) to obtain random sets of disks of varying radii allowing overlap, and smoothen them with a Gaussian kernel. This noise is then added to saturate the intensity values to reflect higher opacity (Fig. 3-c). All augmentations are performed with a probability $p_{aug}$. Additional visualizations of augmented input data are shown in Section 6.2 in the Appendix.

## 3. Data and Experiments

We use publicly available CXR datasets with lung masks – from Shenzhen and Montgomery hospitals – curated for tuberculosis detection (Jaeger et al., 2014) [†]. We use 528 CXRs for training and 176 for validation purposes. These

---

[†] https://www.kaggle.com/kmader/pulmonary-chest-xray-abnormalities

---

datasets do not contain extreme opacification (Fig. 3-a) when compared to cases with opacification (Fig. 3-d). We pool 30 diverse CXRs with high opacification to create a *test* set from different public repositories being curated in response to developing methods useful in detecting COVID-19 (Cohen et al., 2020; Pereira et al., 2020) and a relevant pneumonia detection dataset (Irvin et al., 2019). As these test set images did not have lung masks, we obtained lung masks from expert annotators which are used to validate the proposed method.

We use a U-net (Ronneberger et al., 2015) with modifications as the baseline method which operates at four resolutions, kernel size 3 and has an initial feature map of 24 which are doubled with each of the four downsampling operations. To increase the receptive field of the U-net, the first two resolutions are obtained with a scaling factor of 4 and the other two by a factor of 2. The modifications were based on experiments on the training data where we found increasing the receptive field to be beneficial.

The proposed model utilizes a segmentation network like in the baseline U-net, and an additional variational encoder for data imputation. The variational encoder uses an encoder similar to the one in the baseline model and also operates at four resolutions. The variational encoder also utilizes a sequence of 4 1-D convolution layers to transform the 2-D feature maps to predict $\mu_\phi, \sigma_\phi^2$ of the variational density. We use a latent dimension of 8. As the proposed model has an additional encoder, we reduce the initial feature map to 16 when compared to 24 in the baseline to make the models comparable. This results in about 4.2M parameters for the baseline and about 3.3M parameters for the proposed model.

Both models were developed in PyTorch (Paszke et al., 2019), trained with a batch size of 12, learning rate of $10^{-4}$ with Adam optimizer (Kingma & Ba, 2014) for a maximum of 200 epochs on Nvidia-TitanX GPU with 12 GB memory. Convergence was assumed when there was no improvement in validation loss for 20 consecutive epochs. Model with the minimum validation loss is used for testing. We use a high probability for performing data augmentation, $p_{aug} = 0.9$. The average $CO_2$ footprint of *developing* and training the baseline and proposed models is estimated to be 7.3 kg or equivalently about 60 km traveled by a car, measured using Carbontracker (Anthony et al., 2020).

**Pre- and Post-processing**: All input images are all rescaled to 640x512 px and are histogram equalized to improve the contrast in the images. The predicted segmentations are post-processed with connected component analysis to exclude small erroneous regions and binary morphological closing is done to fill any small holes of radius upto 10 pixels in the segmentation. Example visualizations for these steps are shown in Figures 6 and 7 included in the Appendix.

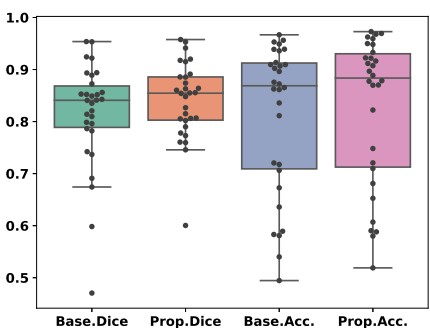

*Figure 4.* Box plot of test set performance for the baseline and proposed models using block masking and diffused noise data augmentations (last two rows of Table 1)

# 4. Results and Discussions

*Table 1.* Performance measures on the test set

| Models | Augmentation | Dice Overlap | Accuracy |
|--------|--------------|--------------|----------|
| Baseline | Standard | $0.7335 \pm 0.17$ | $0.8449 \pm 0.09$ |
| Proposed | Standard | $0.7204 \pm 0.18$ | $0.8392 \pm 0.10$ |
| Baseline | Block | $0.7563 \pm 0.15$ | $0.8522 \pm 0.09$ |
| Proposed | Block | $0.7688 \pm 0.17$ | $0.8552 \pm 0.10$ |
| Baseline | Diffuse | $0.7757 \pm 0.15$ | $0.8654 \pm 0.10$ |
| Proposed | Diffuse | $0.7965 \pm 0.11$ | $0.8652 \pm 0.11$ |
| Baseline | Block+Diffuse | $0.8173 \pm 0.12$ | $0.8654 \pm 0.11$ |
| Proposed | Block+Diffuse | $\mathbf{0.8503 \pm 0.07}$ | $\mathbf{0.8815 \pm 0.11}$ |

We compare the baseline model with the proposed model with variational data imputation in several configurations by varying the data augmentation strategies discussed in Section 2.2. We measure the segmentation performance with two measures: dice overlap and binary accuracy. Results from these experiments are reported in Table 1. Significant performance improvements, based on two-sided paired sample t-tests, when compared to all other configurations are highlighted in bold.

The best dice overlap ($p < 0.05$) and binary accuracy ($p < 0.001$) is obtained by the proposed model with variational data imputation when augmented with block masking and diffused noise, reported in the last row of Table 1. Box plots with performance measures comprising all 30 test set images for the two models used with block masking and diffused noise augmentations are shown in Figure 4. Predicted segmentations for three test set images are visualized in Figure 5 along with the ground truth annotations.

The reported numbers in Table 1 indicate the usefulness of using data augmentation and the use of variational data imputation in a consistent manner. We observe improvements in performance of the baseline model with increasing complexity of data augmentations, in the order listed in Table 1.

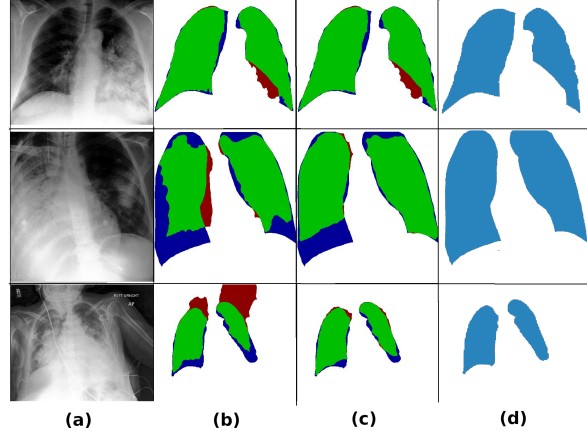

*Figure 5.* a) Three test set samples with highest and least dice accuracy for both methods (rows 1 & 2) along with an input CXR with additional variations in pose (row-3). b) baseline model predictions, c) proposed model predictions and d) the ground truth. Both predictions are for models trained with block and diffused noise.
Green:True positive, Blue: False Negative, Red: False Positive.

With standard augmentation the baseline model obtains a dice accuracy of $0.7335$ which improves to $0.8173$ when using the block masking and diffused noise based augmentations. This trend is also noticed for the proposed model which shows a dice overlap improvement from $0.7204$ to $0.8503$.

Further, the proposed model with variational data imputation shows improvements within each proposed data augmentation category when compared to the baseline method. The lowest p-value was obtained with the block masking and diffused noise reported in the last two rows. This aligns with the hypothesis that the variational data imputation is more effective in learning representations that can handle missing data. The *stochastic* variational block can be thought of as a data dependent noise model that perturbs the learnt feature maps of the U-net encoder, making it robust to missing data, similar to denoising auto-encoders which inject noise to the data to learn useful representations (Vincent et al., 2008).

The qualitative examples shown in Figure 5 show the proposed model is able to output more complete segmentations when compared to the baseline. The predictions in second row show a case of incomplete segmentation predicted by the proposed model for a case with severe opacity. However, the shape of the lungs in this prediction is largely correct when compared to the baseline. This is another consequence of using the variational latent representation as it could help to learn useful features of the desired outputs, which in this case is to predict shapes that are close to lungs. This can also be interpreted as a form of shape regularisation (Esser et al., 2018).

# 5. Conclusions

Several high quality datasets comprising normal CXRs with expert segmentations are publicly available to train segmentation models. However, models trained solely on these data do not generalize well when new data with diverse variations, either due to acquisition or disease, are encountered. We treat such variations as instances of incomplete data and proposed to impute such missing information using the latent representations obtained using a variational encoder. Our trained model which is publicly available now is being used on COVID-19 datasets to obtain lung masks (Cohen, 2020). The quality of the segmentations obtained with this method has been judged to be sufficient and we aim to use them to score COVID-19 risk from CXRs.

## Acknowledgements

The authors would like to thank Anna Zhigalova, Evelina Seduikyte and Joshua Szanyi for their effort in annotating the test set samples. We also thank Abraham Smith and Jens Petersen for useful discussions.

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

# 6. Appendix

## 6.1. Pre- and Post-processing

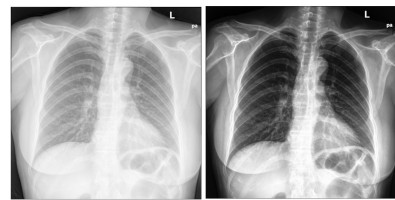

*Figure 6.* Input images a) before and b) after histogram equalization

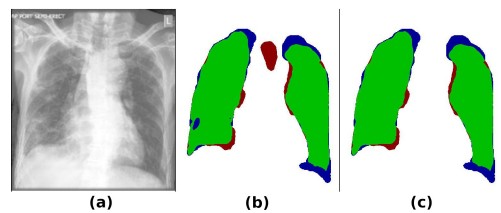

*Figure 7.* a) Input image b) Predicted segmentation overlaid with reference c) Post-processed prediction. Notice the removal of the false positive in the center and closing of a hole in the lower right lung.

## 6.2. Validation visualization

Prediction for some validation set images at convergence for the proposed model with variational data imputation when trained with different augmentations.

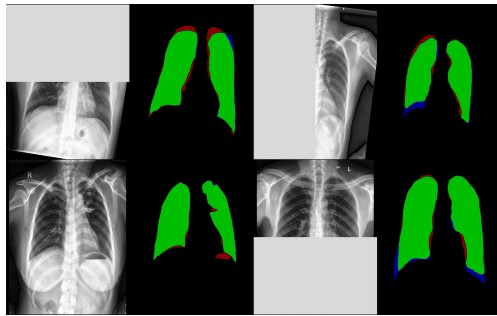

*Figure 8.* Block masking augmentation

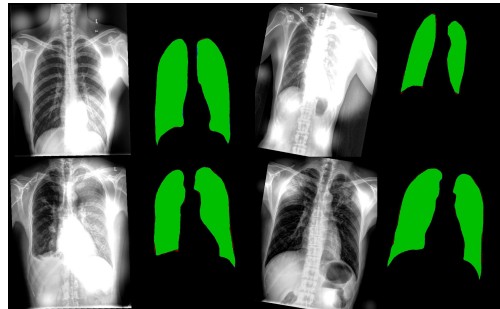

*Figure 9.* Diffused noise augmentation

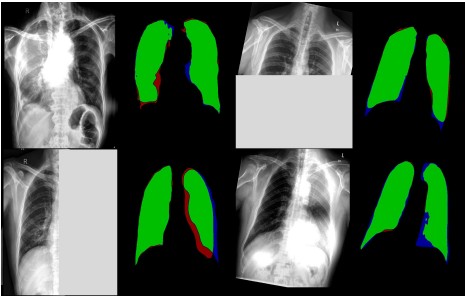

*Figure 10.* Block masking and diffused noise augmentation