# OpenReview forum: "Lung Segmentation from Chest X-rays using Variational Data Imputation"
_ICML.cc/2020/Workshop/Artemiss — ICML Artemiss 2020_

### Official Review · AnonReviewer2 · 2020-06-22
**Interesting idea; could use more clarity on what latent representations are in context of imputation; In line with workshop goals**

**Rating:** 7
**Confidence:** 3

**Review:**

Summary: The authors present a common challenge encountered when working with chest x-rays which is that some x-rays can be high opacity (lower data quality) which can degrade performance. Past work has proposed methods leveraging data augmentation. The authors instead approach this problem as an imputation problem. They use a variational encoder to embed the normal x-rays at train time and concatenate the samples from the encoder inside a conventional U-net architecture for segmentation. They train on publicly available data and test on a curated dataset of opaque images from different datasets.

Strengths:
- The problem is well-motivated given the current pandemic
- The idea is an interesting one and the paper is clearly written

Weaknesses:
- The experiments are only done on 30 images which is a pretty small sample size resulting in large error bars
- It’s not clear to me that the latent distribution is learning a representation of the imputed data (the objective doesn’t seem to directly encourage this)
- Another baseline to compare against might be a supervised/semi-supervised VAE (in this way you can ensure the latent representation is truly reflective of an imputed version of the input)
- Though significant, effect sizes are fairly small

Remarks:
- The main idea seems to be learning a smoother distribution of the inputs — perhaps this can be attempted via a two-step approach to first impute using a generative approach and then segment via another network, this may also be trained end-to-end
- The advantage of this approach would be that you can more transparently evaluate imputation quality


Overall, an interesting idea that can be further developed. This paper is in line with the goals of the workshop though perhaps more emphasis could be placed on the imputation and learned representations in the paper.

---

### Official Review · AnonReviewer1 · 2020-06-23
**clear objective and model structure; good idea**

**Confidence:** 4
**Rating:** 7

**Review:**

The introduction of this paper is well elaborated and can help those who are unfamiliar with this field to understand it concisely. The objective of this artivle and the structure of their model are very clear.

In this work, the VAE is not used as an autoencoder but as a method to perform cross-domain mapping, which seems interesting.

The resluts in a small dataset are good, maybe can use more large datasets.

It is glad to see that the trained model which is publicly available now is being used on COVID-19 datasets to obtain lung masks.

Hope to see more further developped details in the future.

---

### Decision · Program_Chairs · 2020-07-02

**Decision:**

Accept

**Comment:**

We're happy to accept this paper at Artemiss. We'll contact you soon to inform you about more details concerning the format of your presentation at the workshop, and the camera-ready version deadline. Please take into account the referee's comments to write the camera-ready version.